# Clinical Investigation of Chemotherapeutic Resistance and miRNA Expressions in Head and Neck Cancers: A Thorough PRISMA Compliant Systematic Review and Comprehensive Meta-Analysis

**DOI:** 10.3390/genes13122325

**Published:** 2022-12-10

**Authors:** Rama Jayaraj, Karthikbinu Polpaya, Milind Kunale, Gothandam Kodiveri Muthukaliannan, Sameep Shetty, Siddhartha Baxi, Ravishankar Ram Mani, Chitraabaanu Paranjothy, Vinosh Purushothaman, Saminathan Kayarohanam, Ashok Kumar Janakiraman, Ashok Kumar Balaraman

**Affiliations:** 1Jindal Institute of Behavioral Sciences (JIBS), Jindal Global Institution of Eminence Deemed to Be University, Sonipat 131001, India; 2Director of Clinical Sciences, Northern Territory Institute of Research and Training, Darwin, NT 0909, Australia; 3School of Biosciences and Technology, Vellore Institute of Technology (VIT), Vellore 632014, India; 4Department of Oral and Maxillofacial Surgery, Manipal College of Dental Sciences, Mangalore, Manipal Academy of Higher Education, A Constituent of MAHE, Manipal 576104, India; 5MBBS, FRANZCR GAICD (Siddhartha Baxi), Genesis Care Gold Coast Radiation Oncologist, Tugun, QLD 4224, Australia; 6Department of Pharmaceutical Biology, Faculty of Pharmaceutical Sciences, UCSI University Kuala Lumpur (South Wing), No. 1, Jalan Menara Gading, UCSI Heights Cheras, Kuala Lumpur 56000, Malaysia; 7Department of Health, University of Essex, Leeds LS1 2RP, UK; 8Faculty of Health and Life Sciences, INTI International University, Nilai 71800, Malaysia; 9Faculty of Bioeconomics and Health Sciences, University Geomatika Malaysia, Kuala Lumpur 54200, Malaysia; 10Faculty of Pharmacy, MAHSA University, Bandar Saujana Putra, Jenjarom 42610, Malaysia

**Keywords:** head and neck cancer (HNC), miRNA, prognosis, chemoresistance, protocol, systematic review, hazard ratio, patient survival, up-regulation, down-regulation, PRISMA

## Abstract

**Background**: Chemoresistance is a significant barrier to combating head and neck cancer, and decoding this resistance can widen the therapeutic application of such chemotherapeutic drugs. This systematic review and meta-analysis explores the influence of microRNA (miRNA) expressions on chemoresistance in head and neck cancers (HNC). The objective is to evaluate the theragnostic effects of microRNA expressions on chemoresistance in HNC patients and investigate the utility of miRNAs as biomarkers and avenues for new therapeutic targets. **Methods**: We performed a comprehensive bibliographic search that included the SCOPUS, PubMed, and Science Direct bibliographic databases. These searches conformed to a predefined set of search strategies. Following the PRISMA guidelines, inclusion and exclusion criteria were framed upon completing the literature search. The data items extracted were tabulated and collated in MS Excel. This spreadsheet was used to determine the effect size estimation for the theragnostic effects of miRNA expressions on chemoresistance in HNC, the hazard ratio (HR), and 95% confidence intervals (95% CI). The comprehensive meta-analysis was performed using the random effects model. Heterogeneity among the data collected was assessed using the Q test, Tau^2^, I^2^, and Z measures. Publication bias of the included studies was checked using the Egger’s bias indicator test, Orwin and classic fail-safe N test, Begg and Mazumdar rank collection test, and Duval and Tweedie’s trim and fill methods. **Results:** After collating the data from 23 studies, dysregulation of 34 miRNAs was observed in 2189 people. These data were gathered from 23 studies. Out of the 34 miRNAs considered, 22 were up-regulated, while 12 were down-regulated. The TaqMan transcription kits were the most used miRNA profiling platform, and miR-200c was seen to have a mixed dysregulation. We measured the overall pooled effect estimate of HR to be 1.516 for the various analyzed miRNA at a 95% confidence interval of 1.303–1.765, with a significant *p*-value. The null hypothesis test’s Z value was 5.377, and the *p*-value was correspondingly noted to be less than 0.0001. This outcome indicates that the risk of death is determined to be higher in up-regulated groups than in down-regulated groups. Among the 34 miRNAs that were investigated, seven miRNAs were associated with an improved prognosis, especially with the overexpression of these seven miRNAs (miR15b-5p, miR-548b, miR-519d, miR-1278, miR-145, miR-200c, Hsa- miR139-3p). **Discussion:** The findings reveal that intricate relationships between miRNAs’ expression and chemotherapeutic resistance in HNC are more likely to exist and can be potential therapeutic targets. This review suggests the involvement of specific miRNAs as predictors of chemoresistance and sensitivity in HNC. The examination of the current study results illustrates the significance of miRNA expression as a theragnostic biomarker in medical oncology.

## 1. Introduction

### 1.1. Background

Head and neck cancer (HNC) is the sixth most common type of cancer [1]. The epithelial linings of the upper aero-digestive tract, including the oral cavity, oropharynx, hypopharynx, and larynx, are generally affected. HNC affects around 650,000 patients worldwide and accounts for more than 330,000 deaths annually [2]. Human papillomavirus (HPV)-induced oropharyngeal cancers are on the rise and are predominantly seen in young cohorts who are non-smokers and non-alcoholics [3,4,5,6]. The most common causes seem to be alcohol, smoking, and the high risk HPV variants [7,8]. This association is particularly the case for Type 16 (also known as HPV-16) and also occurs with Epstein-Barr viruses [9], which arise from the crypt epithelium of the palatine and lingual tonsils [10]. The standard form of treatment for this form of cancer includes radiotherapy, chemotherapy, or concurrent chemo/radiotherapy [9,11]. Chemotherapeutic drugs such as docetaxel, paclitaxel, and cisplatin treat HNCs [12].

### 1.2. Epidemiology

Worldwide, head and neck carcinoma contributes to more than 650,000 cases and 330,000 deaths every year and ranks as the sixth most common cancer globally [2,13]. In the United States, head and neck carcinomas represent about three percent of malignancies, with roughly 53,000 Americans developing HNC yearly and 10,800 deaths due to this disease. In Europe, there were approximately 250,000 cases (an expected four percent of the disease frequency) and 63,500 deaths in 2012. Males have an increased propensity to this disease with a male–female ratio ranging from 2:1 to 4:1.

The frequency rate in males surpasses 20 per 100,000 in France, Hong Kong, the Indian subcontinent, Central and Eastern Europe, Spain, Italy, Brazil, and African Americans in the United States. Oral cancers are more prevalent in India, and oropharyngeal cancers are more common in the western population [14,15]. The mortality of both laryngeal and oropharyngeal carcinoma is higher in African American men, reflecting the lower prevalence of human papillomavirus (HPV) positivity [16]. The chronic exposure of the upper aero-digestive tract to cancer-causing components such as tobacco use, alcohol consumption, and HPV can bring about dysplastic or premalignant sores/lesions in the oropharyngeal mucosa, which ultimately results in head and neck carcinoma [7,8,17,18,19,20]. The relative frequency of these risk factors adds to the variations in the observed distribution of HNC in various world zones [7].

Studies have shown that the dysregulation of miRNA plays a critical role in cancer progression and contributes to chemotherapeutic resistance or chemoresistance. Preclinical and clinical observational studies have revealed that miRNA expression profiling could improve the classification of high-risk patients with cancer who may develop chemoresistance [21]. The miRNA does so by targeting specific genes/pathways and inhibiting or accelerating those genes’ expression. For example, miR 200-b, miR 155, and miR 146-a, miR 422-a affect the multidrug resistance gene-1 (MDR-1) [22] and causes resistance to CDDP-CRTX and MMC-CRTX. Another study by Bonnin et al. [23] showed that multiple genes and pathways were affected, such as FOXG1, CD73/NT5E oncogene, adenosine receptor-dependent signaling overexpressing CD73.

### 1.3. Rationale

The data on the correlations between HNC chemoresistance/sensitivity and miRNA expression have currently not yielded clinically relevant solutions in the form of theragnostic biomarkers, regardless of the ongoing research in this field. Most of the publications on miRNA-specific chemoresistance of HNC are quite appropriate to the effects of particular miRNA [24,25,26,27,28,29]. The published studies were categorized into samples collected in a hospital for a specific region. After a detailed evaluation of the published literature globally, this systematic review was proffered. This systematic review and meta-analysis provides qualitative and quantitative data on miRNA and methodically assesses the pattern of specific chemoresistance in HNC. This clinical research team previously highlighted a systematic review and meta-analysis approach that permits us to collect the data across all published studies and possibly focus on the associated miRNAs, which have clinical relevance in decisions regarding chemotherapy in patients [21,30,31].

This systematic review and meta-analysis aims to assist researchers and clinicians by quantifying miRNA alterations associated with the chemotherapeutic response in HNC. Forthcoming studies can then identify their utility as predictors of chemotherapy response or theragnosis. This study collates the data on miRNAs and how their regulation can influence the sensitivity of chemotherapeutic drugs.

#### Objectives

The primary objective is to qualitatively analyze the theragnostic effects of miRNA expressions in HNC patients across the world. The secondary objective of this proposed study is to evaluate the up- and down-regulation of miRNAs and evaluate the pooled estimated effect size on the prognosis of HNC patients and resistance in cell lines that may cause recurrence.

## 2. Search Strategy and Methods

The study was conducted by the Preferred Reporting Items for Systematic Reviews and Meta-Analyses (PRISMA) statement [32] and was completed following a previously established protocol (PROSPERO registration number: CRD42018104657). The study protocol was already published elsewhere [33].

### 2.1. Review Questions

What effect does miRNA regulation have on chemotherapy?

What is the general prognosis of patients having miRNA-specific chemoresistance?

What are the miRNAs most responsible for chemoresistance in HNC patients?

What are the survival rates associated with each miRNA linked to chemoresistance, and how are they affected?

### 2.2. Study Design

#### Search Strategy

The PubMed and Science Direct databases were searched for publications published between 2008 and 2021. The Medical Subjective Heading (MeSH) search phrases were used in the search (Table 1). There were no limits on study participants regarding age, gender, ethnicity, country of origin, and morbidities (for patients and the general population). Four authors of this study (RJ, MRM, PS, and MR) independently assessed the titles and abstracts to see if the publications satisfied the inclusion criteria. In accordance with the protocol:

The selected full-text papers were checked for studies that did not include abstracts.

The reference lists of the collected studies were manually searched to improve the robustness of the search results.

The cross-references from the selected studies were searched for additional articles.

When the relevant information was not available in the publication, we contacted the corresponding authors.

Any discrepancies were resolved through discussion and consensus with a third reviewer.

### 2.3. Selection Criteria

#### 2.3.1. Inclusion Criteria

Studies analyzing the theragnostic effects of miRNA expressions in both HNC patients and cell lines were considered.

Studies analyzing miRNAs and chemoresistance/are performed in liquid biopsies (of plasma and saliva samples) were included.

Studies that discussed HNC patients’ clinicopathological characteristics and the hazard ratio (HR) or Kaplan–Meier curve were included.

Articles that discussed the survival outcomes of almost all stages of HNC patients were included in the meta-analysis.

Studies reporting miRNA profiling platform and miRNA expressions analysis using in vitro assays were included.

Genes and pathways involved in chemoresistance or sensitivity in HNC patients were also considered.

Studies appropriate to PRISMA guidelines for systematic review and meta-analysis were included.

#### 2.3.2. Exclusion Criteria

Studies published in languages other than English were excluded.

Any information or results from letters to the editors, case studies, conference abstracts, case reports, and review articles of HNC were removed.

Studies performed only in patients or in vitro were excluded and were not considered for the systematic review.

Studies lacking proper discussion about miRNA profiling and pathways related to that were excluded.

Studies with no accessibility to survival outcomes, HR, or Kaplan–Meier (KM) curves were not considered for the meta-analysis.

Studies whose full texts were not accessible were excluded.

Duplicates were removed, and the study was excluded if it fell within the exclusion criteria.

### 2.4. Data Extraction and Management

All studies that satisfied the selection criteria were assessed, and all clinical and histological parameters for patients were extracted. Author names, year of publication, study location, study period, gender, sample size, source of a clinical sample, miRNAs profiling platform, follow-up period, miRNAs studied, histological type, lymph node metastasis/distant metastasis, clinical stages, and survival data were all sorted under the following headings: author names, year of publication, study location, study period, gender, sample size, source of a clinical sample, miRNAs profiling platform. The data from the studies that qualified for final inclusion were tabulated using a Microsoft Excel spreadsheet.

### 2.5. Assessment of Quality

The study quality of the literature extracted for the systematic review and meta-analyses were assessed using the Meta-Analysis of Observational Studies in Epidemiology (MOOSE) checklist. Studies that satisfied 0–33% of the 14 items on the checklist were considered to be of “poor” quality, with “satisfactory” study quality indicating an adherence of 33–66% of the study to the checklist, while “good” quality studies were in the range of 67–100% adherence. All studies included fell within either satisfactory or good study quality. The items specified in the MOOSE checklist are delineated in Table 2.

### 2.6. Publication Bias

Publications bias indicators of the included studies were assessed using Orwin and classic fail-safe N test [34], Egger’s bias indicator test, Begg and Mazumdar Rank collection test, Duval and Tweedie’s trim fill calculation [35,36], and inverted funnel plot.

### 2.7. Comprehensive Meta-Analysis

Comprehensive meta-analysis was performed to estimate the pooled estimated effect size HR and 95% CI from the included studies using comprehensive meta-analysis (CMA) software version 3.0. Random effects models were used for meta-analysis. Cochran’s Q test, Tau square, Z value, and I^2^ statistic [37,38] were performed to assess the heterogeneity and hypothesis testing of the included studies. The random effects model was performed when the *p*-value > 0.05, and heterogeneity was observed. A forest plot was drawn to summarize the pooled HR estimate of the chemoresistance-specific miRNAs.

## 3. Results

### 3.1. Study Selection:

The selected and eligible studies for this systematic review and comprehensive meta-analysis through search results are shown in the flow chart in Figure 1. Studies were searched using critical terms, as seen in the protocol paper. Upward of 4610 records appeared upon merely searching MeSH keywords. After searching the duplicate records and records marked as ineligible, we had 459 papers. After shortlisting by scanning the titles and abstracts for relevant papers, we had 113 articles. After shortlisting the papers that did not match the selection criteria, we had 34 studies, out of which 23 were used for the meta-analysis as they had the required data. All these studies underwent quality assessment using the MOOSE checklist and were of acceptable quality for inclusion in a meta-analysis study.

### 3.2. Study Characteristics

The results for the various parameters analyzed in this systematic review are shown in Table 3. As seen in the table, most of the studies were from China, Germany, and Japan. Thirty-four miRNAs were analyzed from a patient population of 2189 people. Tissue and serum samples were the most used in the studies analyzed in this review. In the studies analyzed, cisplatin was the most commonly used chemotherapeutic drug for which chemoresistance was observed. Out of the total 34 miRNAs, 22 miRNAs were up-regulated, while 12 miRNAs were down-regulated. The TaqMan transcription kit was the most commonly used of the miRNA profiling platforms. miRNA 200c was seen to have a mixed dysregulation; it was seen to be up-regulated in a study by Hamano R [39] and was seen to be down-regulated in a study by Song J [40].

#### 3.2.1. In Vitro Assays

This section illustrates the commonly used in vitro assays collected from the studies represented by the following Figure 2 and Figure 3 below. In the 24 studies utilized in this review, 19 cell lines were used, OECM-1. Of these, the SAS cell lines were the most commonly used (Figure 3). Of the data collected from all the studies, the highest number of cell lines used in a single study was eight. Among the data collected, we also analyzed certain in vitro studies used in the collected studies. The most commonly used assays included qRT-PCR, cell proliferation assay, MTT assay for cell viability and cytotoxicity, luciferase reporter assays, Western blotting, apoptosis assay, clonogenic assay, scramble assay, immunoprecipitation as well as immunohistochemistry assays, RFLP assay, electrophoretic mobility shift assay, chemosensitivity, chromatin immunoprecipitation (ChIP) assay among others. Figure 2 summarizes how frequently the most common assays were used in the studies considered.

#### 3.2.2. Relation between miRNA Expression and Chemoresistance

The following data were obtained from the collected data analysis of the results. Out of the 34 miRNAs investigated in the study, the seven miRNAs (miR15b-5p, miR-548b, miR-519d, miR-1278, miR-145, miR-200c, Hsa- miR139-3p) were linked to better survival, while the rest of the 27 miRNAs were associated with poor survival. The following nine miRNAs were known to affect chemoresistance in HNC miR-200, miR-34a, miR-196a, miR-27a, miR-27b, miR-200c, miR-494, miR-1290, and miR-205. These mentioned miRNAs are up-regulated, except miR-34a, which is down-regulated. The following three miRNAs, miR-519d, miR-1278, and miR-29c, are known to inhibit chemoresistance and are noted to be down-regulated. The most commonly used chemotherapy drug among the nine drugs is cisplatin. Overall, 13 miRNAs were associated with regulating chemoresistance to chemotherapy drugs, as well as certain miRNAs such as miR-1290, which are known to affect the commonly used chemoradiotherapy (CRT).

#### 3.2.3. Chemotherapy and HNC Patients

There were a total of nine drugs used as chemotherapy in the pooled studies: cisplatin (866 patients), 5-fluorouracil (646 patients), doxorubicin (260 patients), paclitaxel (151 patients), cetuximab (152 patients), oxaliplatin (33 patients), leucovorin (33 patients), silibinin (45 patients), and docetaxel (97 patients).

#### 3.2.4. Drug Regulatory Pathways for miRNA-Mediated Chemosensitivity and Chemoresistance

Figure 4 below represents the various pathways affected in HNC and comprehensively illustrates the deregulation of miRNA. From the articles included in the study, 11 pathways and their associated genes were investigated and elaborated on in individual studies. Four pathways were described as associated with cell survival. In contrast, two pathways were related to apoptosis, four pathways were analyzed to be linked with cell differentiation and proliferation, while one was involved in angiogenesis.

#### 3.2.5. Association between miRNAs and Drug Regulatory Pathways of Chemoresistance

From the studies analyzed, it was noted that many miRNAs were up-regulated when chemoresistance was observed compared to the number of down-regulated miRNA. Chemoresistance to cetuximab was marked by up-regulation of nine miRNA, namely, miR-15b-5p, miR-92a, miR-548b, miR-103, miR-18a, miR-205, miR-532, miR-20a, and miR-365. Of these, miR 15b-5p affected p16, EGFR, and CD44; miR-15b-5p/TRIM-29/PTEN/AKT/mTOR signaling pathways; and others affected the PI3K/Akt/mTOR pathways. miR-218, Let-7g, and miR-125b were down-regulated during M_4_N treatment, and the Hedgehog and Wnt signaling cascades, SP1, MYC, and TP53 genes were seen to be affected. Up-regulation of miR-200b in CDDP-CRTX chemoresistance and up-regulation of miR-155 along with miR-146 in MMC-CRTX chemoresistance were all seen to affect the multidrug resistance gene-1. Chemoresistance towards silibinin, doxorubicin or cisplatin, or fluorouracil was marked by up-regulation of miRNA-494, which affected ZEB2 and β-catenin signaling, ADAM10, FOXM1, CD44, and ALDH1 pathways, and also Bmi1/ADAM10 pathways (Table 4).

miRNA 200c was seen to have mixed dysregulation, up-regulation of miR-200c was seen in the chemoresistance towards cisplatin, and the Akt pathway, MDR-1 gene, and PPP2R1B gene pathway were seen to be affected, while in another study, down-regulation of miR-200c was noted to affect the TP53 pathway.

### 3.3. Comprehensive Meta-Analysis

We analyzed the dysregulation of 34 miRNAs seen in 2189 HNC patients. These data were gathered from 23 studies. This analysis revealed that out of 34 miRNAs, 22 were up-regulated, while 12 were down-regulated. miRNA 200c was observed to be up-regulated in the Akt pathway of patients undergoing cisplatin treatment. In contrast, the same miRNA was seen to be down-regulated in the TP53 pathway in a different study. The overall pooled effect estimate for the various miRNAs was 1.516 with a 95% confidence interval of 1.303–1.765. The HR and the 95% CI for the studies included in this paper utilizing the fixed effect model are 1.302 (1.222–1.386). Moreover, the adjusted point estimate and the 95% CI values with the fixed model as reference are 1.263 (1.192–1.349). The HR and 95% CI values utilizing the random effects model are 1.516 (1.303–1.765). The point estimate and the 95% CI values with the random effects model are 1.336 (1.151–1.552). Table 5 depicts the publication bias indicators hypothesis testing and heterogeneity testing analysis of miRNA-specific chemoresistance in HNC.

### 3.4. Influence of miRNA Expression on the Survival of HNC Patients

In the results mentioned above, the Z value from the null hypothesis test was 5.377, and the *p*-value was correspondingly noted to be less than 0.0001. This result indicates that the risk of death is determined to be higher in up-regulated groups than in down-regulated groups. Among the 34 miRNAs that were investigated, 7 miRNA (miR15b-5p, miR-548b, miR-519d, miR-1278, miR-145, miR-200c, Hsa- miR139-3p) were associated with an improved prognosis (better survival). The other 27 miRNAs (miR-29c, miR-626, miR-5100, miR-21, miR-200b, miR-365, hsa-miR-194-5p, miR-200c(a), miR-200c(b), miR-532, miR-20a, miR-21, miR-155, miR-27a, miR-21, miR-494, miR-146a, miR-196a, miR 675(a), miR-149, miR-205, miR-18a, miR-103, miR-422a, miR-675(b), Let-7g(a), Let-7g(b), miR-1290(a), miR-1290(b), miR-92(a), miR 27b, miR-218) were associated with a poor prognosis (poor survival).

### 3.5. Extent of Variance of Estimated Effect Size across Included Studies

This study applied the Q-Statistics test, which assumes that all studies used in the analysis have the same impact size. With 39 degrees of freedom (df) and a noted *p*-value of less than 0.0001, the calculated Q value was 118.561. We cannot reject the null hypothesis as the true effect size was similar in all the included studies. In addition, the observed variation falls within the range assigned to the sampling error. The I^2^ statistic referred to the extent of observed variance that can effectively illustrate the differences in effect size instead of sampling error. I^2^ is 67.106% in this study. T^2^ (T = tau) (in log units) effectively denotes the variance of accurate effect sizes. In this study, the T^2^ value is 0.098. T stands for the standard deviation of actual effects (in log units). In this study, the T-value is 0.314.

### 3.6. Publication Bias and Sensitivity Analysis—Funnel Plot

Publication bias as an indicator is used because many studies that are complete are not published due to the outcome of the study, wherein the results may not be significant. A funnel plot was developed (Figure 5), which was asymmetric across survival outcomes. The asymmetry represents the presence of publication bias. The vertical axis represented the study size’s standard error and precision, and the horizontal axis represented the effect size. The dots represent individual studies, and one can appreciate that most of the studies are in the high significance region. This indicates the presence of publication bias.

### 3.7. Orwin’s Fail-Safe N Test

In the studies in this meta-analysis review, the hazard ratio (HR) values were measured to be 1.30155. The mean hazard ratio cited in the results was 1.000 (generally can consist of any value other than a nil value). In this review, the HR observed to be 1.30155 is not placed between the mean HR of the missing studies, which is 1.000 [34].

### 3.8. Begg and Mazumdar Rank Correlation Test

This test is generally performed to correlate Kendall’s Rank with the standardized effects sizes and standard errors. The yield of a positive value in this test is indicative of a high degree of accuracy of the included studies in this meta-analysis. Kendall’s Tau (rank-order correlation) values were found to be 0.15897 (without continuity correction) and 0.15679 (with continuity correction). Subsequently, the *p*-tailed values for 1-tailed and 2-tailed were established as 0.07592 and 0.15184, respectively.

In Figure 6, CMA software was used to calculate and analyze the HR values’ pooled hazard ratios for HNC prognostic data. The meta-analysis was conducted by analyzing 23 studies involving 34 miRNAs from a combined patient pool of 2189 HNC patients. The analysis yielded a Z value of 8.63 with a *p*-value of less than 0.001.

### 3.9. Egger’s Test of Intercept

This study yielded an intercept of 0.89103 at 95% CI (0.14381–1.63825), *t*-value = 2.41400, and 38 degrees of freedom. The *p*-value generated for the one-tailed test was 0.01305, and the *p*-value for the two-tailed test was 0.0207.

### 3.10. Duval and Tweedie’s Trim and Fill Test

This test is instrumental in the study as it helps diminish the effect of publication bias. This test is generally performed when the funnel plot observed is asymmetrical [35]. The studies that contribute to the asymmetry are trimmed from the right side of the funnel plot to pinpoint the unbiased effect. This is then filled back by re-inserting the trimmed studies on the right and the imputed studies on the left side of the mean effect. In this review, approximately ten studies that produced asymmetry in the plot were trimmed and filled. This funnel plot was created using CMA software (Englewood, NJ 07631 USA) and illustrates the trimmed and imputed studies.

## 4. Discussion

Head and neck cancers (HNC) are plagued by the inherent chemoresistance towards the most commonly used drugs in HNC, such as cisplatin and cetuximab. This drug resistance tends to lead to rapid deterioration of the long-term prognosis of the patient [25,57]. This study aims to evaluate the potential role of miRNAs, which are one type among several types of small non-coding RNAs known to play a specific role in cancer progression, including the development of chemoresistance [25,33]. The regulation of chemoresistance specific miRNA is significant in the genesis of cancer as well as in the prognosis of the affected patient. miRNAs are also known to play a significant role in apoptosis, DNA repair, and epithelial–mesenchymal regulation in the cell cycle. The previous meta-analysis on HNC illustrated the role of miRNAs in targeting patient survival [25]. Dai et al. performed an earlier descriptive review on HNC that investigated the role of miRNAs in targeting drug regulatory receptors; however, the authors of this study highlighted only three miRNAs related to chemoresistance in HNC [58]. Hence a systematic review that included a comprehensive meta-analysis of thirty-four miRNAs that impact chemoresistance to drugs in HNC was needed. This systematic review was performed using 459 articles obtained through MeSH PubMed key search terms, among which 34 publications were included for a systematic review, and 23 articles were included for a comprehensive meta-analysis based on selection criteria.

The pathological parameters were evaluated and analyzed to effectively correlate and understand the risk factors that may affect or aggravate the disease progression. The hazard ratio values and the 95% CI values were also collected and tabulated to create forest plots that illustrate the role of each miRNA influencing the patients’ prognosis. These miRNAs showed chemoresistance to malignant cells by silencing or inactivating pathways that promote chemoresistance directly or indirectly. For instance, in a study conducted by Martz et al. [59], activation of certain pathways such as Notch-1, phosphoinositide 3-kinase (PI3K), and mammalian target of rapamycin (mTOR), PI3K/AKT, and estrogen receptor (ER) signaling pathways tend to induce chemoresistance to various drugs used for treatment.

### 4.1. Strengths of the Study

Global literature-based meta-analysis: The studies collected for this systematic review and meta-analysis are abreast with the recent global literature. The impact of certain miRNA on treatment regimens for different HNC patients was looked at from studies collected worldwide. Best research practice in the HNC field: This study adheres to apposite research practice and statistical guidelines. The study’s findings were reported according to the PRISMA guidelines and were registered in PROSPERO.

Clinical recommendation for future studies: This review provides a template for future studies exploring the clinical utility of miRNA.

Methodologically sound analysis: Most of the studies included in this review were of acceptable quality, and the application of quality evaluation tools proved the study’s methodological quality.

Publication bias indicators: A detailed evaluation of publication bias indicators is a fundamental parameter of meta-analysis, which aids any biases in reporting original literature-based meta-analyses of previously published studies. In addition, as per the PRISMA guidelines, an additional investigation of publication bias indicators for small and missing studies was recommended.

First comprehensive meta-analysis study: The authors identify that this is one of the first systematic reviews and meta-analyses on chemoresistance-specific miRNAs in HNC patients.

### 4.2. Limitations of this Study

Despite the retrospective data collated globally, a significant proportion of the included studies arose primarily from China, Canada, Japan, and Germany, limiting the widespread applicability of the studies. In some studies, HR and the 95% confidence interval data were not directly provided and had to be extracted from the Kaplan–Meier Curves, leading to estimation errors. Each study used varying analysis procedures, such as different techniques and sample sources. This leads to inherent heterogeneity between the studies and could contribute to bias.

## 5. Conclusions

This comprehensive review and meta-analysis offer conclusive evidence on the role of the miRNAs that affect the survival of patients by affecting the chemoresistance and the disease progression in patients. The regulation of these miRNA is crucial in terms of prognosis and survival. Using forest plots and other statistical methods, we conclusively cement our findings that certain miRNA may negatively affect the patient’s survival leading to a poor prognosis. Future longitudinal research with patient-based meta-analysis is essential to demonstrate the specific miRNAs that may be intricately involved in chemoresistance in HNC.

## Figures and Tables

**Figure 1 genes-13-02325-f001:**
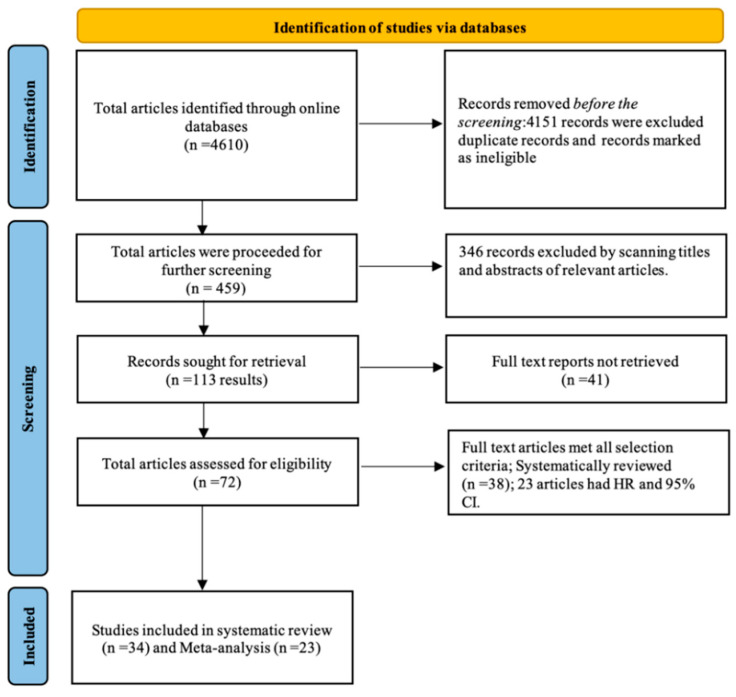
Flow chart of the literature search.

**Figure 2 genes-13-02325-f002:**
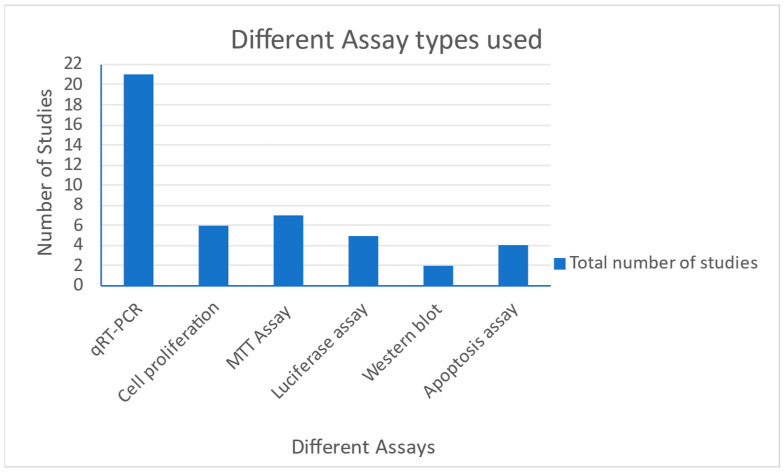
Chart showing the various assays performed in the collected studies.

**Figure 3 genes-13-02325-f003:**
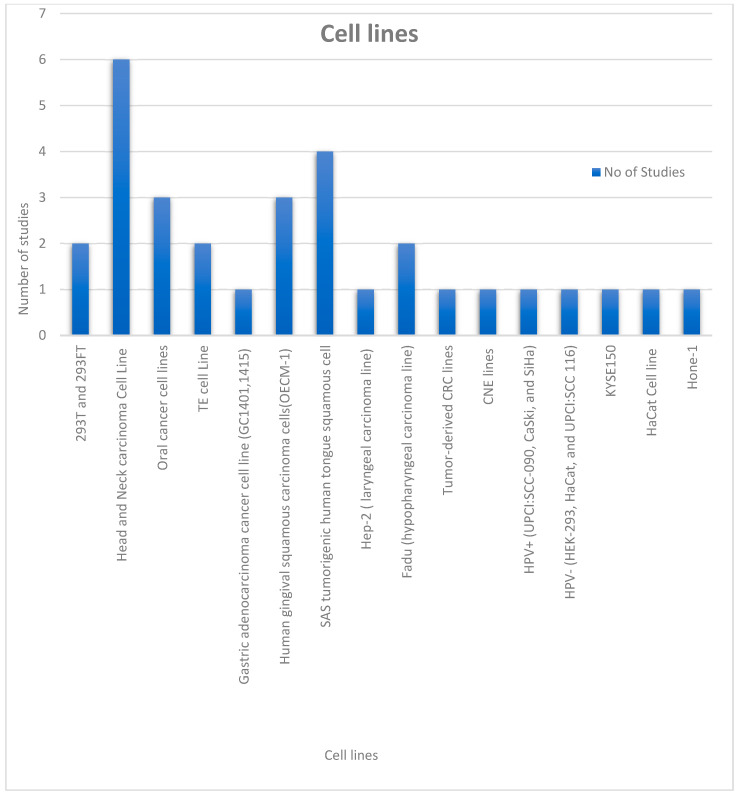
Chart showing the various cell lines utilized in the collected studies.

**Figure 4 genes-13-02325-f004:**
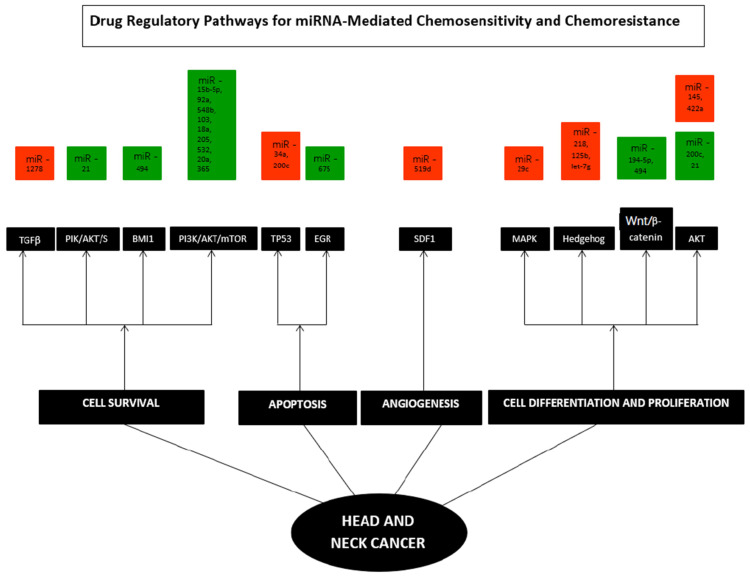
**Hallmarks of specific miRNAs in Head and Neck Cancer.** TGFβ, PIK/AKT/S6, BMI1, PI3K/AKT/mTOR, TP53, EGR, SDF1, MAPK, Hedgehog, Wnt/β- catenin, and AKT pathways. Each hallmark depicts several examples of miRNAs that control specific cellular processes in HNC; some microRNAs influence multiple hallmarks, implying that they govern various pathways. **Orange**—down-regulated miRNA; **Green**—up-regulated miRNA.

**Figure 5 genes-13-02325-f005:**
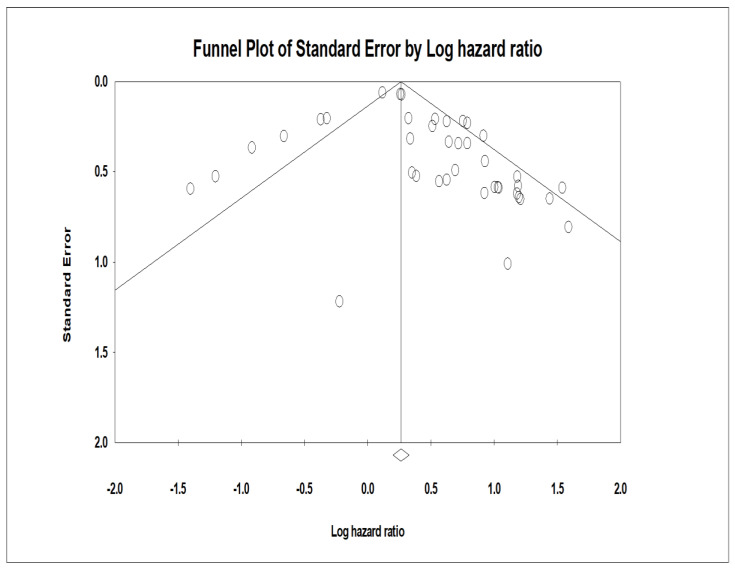
Funnel plot of the studies included in this comprehensive meta-analysis of miRNA.. Funnel plot with inputted studies (studies included in this comprehensive meta-analysis of miRNA-specific chemoresistance in head and neck cancer). The black circles indicate imputed studies.

**Figure 6 genes-13-02325-f006:**
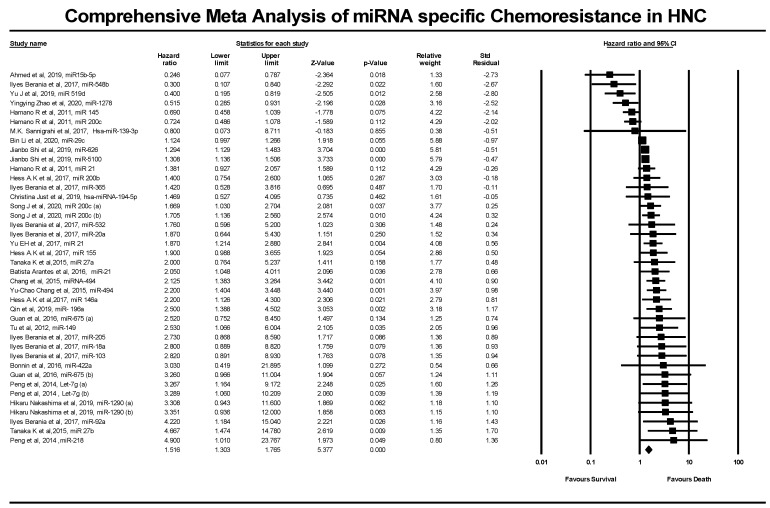
Forest plot of the studies included in this comprehensive meta-analysis of miRNA-specific chemoresistance in head and neck cancer.

**Table 1 genes-13-02325-t001:** Key terms utilized in the search strategy.

S No.	Search Items
1.	“miRNA” [Topic] AND “treatment” [Topic] OR drug resistance” [Topic] AND “HNC” [Topic] OR “Head and Neck Cancer” [Topic]
2.	“microRNA” [Topic] AND “drug-resistance” [Topic] AND “HNC” [Topic] OR “Head and Neck Cancer” [Topic]
3.	“Up-regulation OR down-regulation in HNC” [Topic] OR “Differential Expression” [Topic] OR “Deregulated miRNAs” [Topic] OR “Head and Neck Cancer” [Topic]
4.	“miRNA” [Topic] AND “chemotherapeutic resistance” [Topic] OR “chemosensitivity” [Topic] AND “HNC” [Topic] OR “Head and Neck Cancer” [Topic]
5.	“miRNA” [Topic] AND “treatment resistance” [Topic] OR “chemoresistance” [Topic] AND “HNC” [Topic] OR “Head and Neck Cancer” [Topic]
6.	“microRNA” [Topic] AND “chemosensitivity” [Topic] AND “HNC” [Topic] OR “Head and Neck Cancer” [Topic]
7.	“microRNA” [Topic] AND “chemoresistance” [Topic] AND “HNC” [Topic] OR “Head and Neck Cancer” [Topic]
8.	“HNC survival outcome” [Topic] OR “Hazard Ratio” [Topic] AND “HNC” [Topic] OR “Head and Neck Cancer” [Topic]

**Table 2 genes-13-02325-t002:** The Meta-Analysis of Observational Studies in Epidemiology (MOOSE) checklist.

S. No	Criteria
**1**	The objective of this paper stated
**2**	The study population clearly stated
**3**	Participation rate of eligible persons is at least 50%
**4**	Eligibility criteria
**5**	Sample size justification
**6**	miRNA exposure assessed before outcome measurement
**7**	Timeframe sufficient for the patients (OS, DFS, or MFS)
**8**	Different levels of the exposure of interest (mode of treatment)
**9**	Exposure measures and assessment (staging of cancer, TNM)
**10**	Repeated exposure assessment
**11**	Outcome measures (HR, CI)
**12**	Binding of outcome assessors
**13**	Follow-up rate
**14**	Statistical analysis

**Table 3 genes-13-02325-t003:** Description of the 23 included studies.

S. No	Study	Country	No. of Patients	Sex (M/F)	No. of Samples (Cancer/Normal)	Type of Sample	Chemotherapy	Resistant Cells	Smoking History	Alcohol Consumption	HPV Positive/Negative	Clinical Stages (Old)			Cancer Type/Subtype	Lymph Node Metastasis/Distant Metastasis	Cell Lines	miRNA	miRNA Dysregulation	miRNA Profiling Platform	Pathways/Gene
									Smoker or Ex-Smoker/Non-Smoker	Drinker or Ex Drinker/Non Drinker		Overall Stages	I–II	III–IV							
1	Hess A K et al. (2017) [22]	Germany	149	123/26	149/0	Tissue	CDDP-CRTX	NM	92/55	NM	12/62	TNM Stage (IV)	0	149 (Stage IV)	NM	NM	NM	miR-200b	Up-regulated	GeneChip miRNA 2.0 Array, TaqMan MicroRNA Assays	Multidrug resistance gene 1
2	Hess A K et al. (2017) [22]	Germany	149	123/26	149/0	Tissue	MMC-CRTX	NM	92/55	NM	12/62	TNM Stage (IV)	NM	149 (Stage IV)	NM	NM	NM	miR-155	Up-regulated	GeneChip miRNA 2.0 Array, TaqMan MicroRNA Assays	Multidrug resistance gene 1
3	Hess A K et al. (2017) [22]	Germany	149	123/26	149/0	Tissue	MMC-CRTX	NM	92/55	NM	12/62	TNM Stage (IV)	NM	149 (Stage IV)	NM	NM	NM	miR-146a	Up-regulated	GeneChip miRNA 2.0 Array, TaqMan MicroRNA Assays	Multidrug resistance gene 1
4	Ogawa T et al. (2012) [41]	Japan	24	16/8	24/17	Tissue	CDDP-CRTX	RPMI2650 CR	NM	NM	NM	T2, T3, T4a and N0, N+	1	23	T2, T3, and T4a	N0, N1	RPMI2650	miR-34a	Down-regulated	Human miRNA microarray ver 3, Sanger miRBase Release 12.0, GeneSpring Software, TaqMan MicroRNA Assays	TP 53
5	Yu EH et al. (2017) [42]	RoC	100	92/8	102/0	Tissue	5-fluorouracil	NM	NM	NM	NM	TNM Stage (I, II, III, and IV)	23	77	T1, T2, T3, and T4	N0, N+	NM	miR-21	Up-regulated	miRCURY LNA miR-21 probe, Exigon Scrambled Probe	PI3K/AKT/S6 pathway, PTEN
6	Qin X et al. (2019) [43]	RoC	80	43/37	160/30	Tissue and Serum	Cisplatin, doxorubicin, paclitaxel	HN4-res	30/50	24/56	NM	TNM Stage (I, II, III, and IV)	33	47	NM	N0, N1, N2	SCC-4, SCC-9, SCC-25, CAL 27, 293T, HN4, HN6, and HN30	miR-196a	Up-regulated	Prime Script RT reagent Kit	CDKN1B, ING5
7	Tanaka K et al. (2015) [44]	Japan	64	50/14	64/27	Serum	Cisplatin, docetaxel, 5-fluorouracil	NM	NM	NM	NM	TNM Stage (II, III, and IV)	23	49	T1, T2, T3, and T4	N0, N1/M0, M1	TE10, TE8	miR-27a	Up-regulated	mirVana PARIS kit, TaqMan Array Human MicroRNA Assay kit	FOXO1, MET, MDR-1 Gene
8	Tanaka K et al. (2015) [44]	Japan	64	50/14	64/27	Serum	Cisplatin, docetaxel, 5-fluorouracil	NM	NM	NM	NM	TNM Stage (II, III, and IV)	23	49	T1, T2, T3, and T4	N0, N1/M0, M1	TE10, TE8	miR-27b	Up-regulated	mirVana PARIS kit, TaqMan Array Human MicroRNA Assay kit	FOXO1, MET, MDR-1 Gene
9	Hamano R et al. (2011) [39]	Japan	98	84/14	98/0	Tissue	Cisplatin	TE-8R	NM	NM	NM	TNM Stage (I, II, III, and IV)	34	64	T0, T1a, T1b, T2, T3 and T4	N0, N1	TE-1, TE-8, TE-10, TE-13, TE-15	miR-200c	Up-regulated	TaqMan miRNA transcription kit, TaqMan Universal PCR Master Mix	Akt pathway, MDR-1 gene, PPP2R1B Gene
10	Hamano R et al. (2011) [39]	Japan	98	84/14	98/0	Tissue	Cisplatin	TE-8R	NM	NM	NM	TNM Stage (I, II, III, and IV)	34	64	T0, T1a, T1b, T2, T3 and T4	N0, N1	TE-1, TE-8, TE-10, TE-13, TE-15	miR-145	Down-regulated	TaqMan miRNA transcription kit, TaqMan Universal PCR Master Mix	Akt pathway, MDR-1 gene, PPP2R1B Gene
11	Hamano et al. (2011) [39]	Japan	98	84/14	98/0	Tissue	Cisplatin	TE-8R	NM	NM	NM	TNM Stage (I, II, III, and IV)	34	64	T0, T1a, T1b, T2, T3 and T4	N0, N1/M0, M1	TE-1, TE-8, TE-10, TE-13, TE-15	miR-21	Up-regulated	TaqMan miRNA transcription kit, TaqMan Universal PCR Master Mix	Akt pathway, MDR-1 gene, PPP2R1B Gene
12	Song et al. (2020) [40]	China	204	146/58	204/0	Tissue	NM	NM	133/71	168/36	NM	TNM Stage (I, II, III, and IV)	113	91	T1, T2, T3, and T4	N+	HOC 313	miR-200c	Down-regulated	TaqMan™ MicroRNA Reverse Transcription Kit, TaqMan™ MicroRNA Assays	TP 53
13	Yu J et al. (2019) [45]	China	60	41/19	120/0	Tissue	NM	NM	28/32	22/38	NM	TNM Stage (I, II, III, and IV)	22	38	NM	N0, N1, N2	HN4 and HN30	miR-519d	Down-regulated	PrimeScript™ RT reagent kit, miRcute Plus miRNA First-Strand cDNA Synthesis Kit, ABI StepOne Real-Time PCR System, SYBR Premix Ex Taq Reagent Kit	CXCR4
14	Ahmed et al. (2019) [9]	Czech Republic	94	94/0	43/0	Tissue	Cetuximab	NM	NM	NM	NM	TNM stages (I, II, III, and IV)	1	42	T1a, T1b	NM	Oral cancer cell lines (ACOSC3,ACOSC4)	miR-15b-5p	Up-regulated	QuantStudio 12K Flex Real-Time PCR System following TaqMan MicroRNA Assay	p16, EGFR, and CD44; miR-15b-5p/TRIM-29/PTEN/AKT/mTOR signaling pathway
15	Christina Just et al. (2019) [46]	Germany	33	26/7	21/12	Tissue	5-fluorouracil, leucovorin, oxaliplatin, and docetaxel	NM	NM	NM	NM	NM	NM	NM	T0, T1a, T1b, T2, T3, T4a, T4b	N0, N1, N2, N3/studied but not mentioned	Esophagogastric cancer cell lines (GC1401, GC1415)	miRNA-194-5p	Up-regulated	PCR using LightCycler^®^ 480 Software (Roche molecular systems Inc., Mannheim, Germany)	PTEN, BCL2, IGF1R, Wnt/β-catenin pathway; DKK2, CDH1, CD44, MYC, and ABCG2 expression
16	Chang et al. (2015) [11]	Taiwan	45	NM	45/0	Tissue	Silibinin, doxorubicin cisplatin, or fluorouracil	ALDH1, CD44, and HNC- TICs	NM	NM	NM	TNM stages (I, II, III, and IV)	NM	NM	T0, T1	N0	Human gingival squamous carcinoma cells (OECM-1); SAS tumorigenic human tongue squamous cell	miRNA-494	Up-regulated	TaqMan miRNA assays with specific primer sets (Applied Biosystems, Carlsbad, CA, USA)	ZEB2 and β-catenin signaling, ADAM10, FOXM1, CD44, and ALDH1
17	Bonnin et al. (2016) [23]	France	75	61/14	36/39	Tissue	RT and RT-CT	NM	59/6	56/6	10/65	CS (III and IV)	NM	75	T3, T4	N0, N1, N2, N3	SCC61, SQ20B (HNSC), and HaCaT (normal)	miR-422a	Down-regulated	TaqMan^®^ MicroRNA Assays and MxPro 3000(Agilent, St. Clara, CA, USA); QuantiTect SYBR^®^ Green PCR Kit (Thermo Fischer Scientific, Waltham, MA, USA)	FOXG1, CD73/NT5E oncogene, adenosine receptor-dependent signaling overexpressing CD73
18	Batista Arantes et al. (2016) [47]	Brazil	71	68/3	47/0	Tissue	cisplatin and paclitaxel	NM	57/14	27/44	6/65	CS (III and IV)	NM	71	T2, T3, T4	N0, N1, N2, N3	OCSS (oral squamous cell carcinoma)	miR-21	Up-regulated	TaqMan PCR kit on 96-well plates in the 7900HT Akt pathwayFast Real-Time PCR System (Applied Biosystems)	
19	Guan et al. (2016) [48]	China	62	48/14	62/0	Tissue	NM	Hep-2 cells	NM	NM	NM	TNM Stage (I–II and III–IV)	18	44	T1, T2, T3, T4	studied but not mentioned	Hep-2 (laryngeal carcinoma line) and Fadu (hypopharyngeal carcinoma line)	miR-675	Up-regulated	qRT-PCR analysis using SYBR Green Master Mix (Applied Biosystems) and ABI PRISM 7900 Sequence Detection System (Applied Biosystems Inc., Foster City, CA, USA)	Wnt signaling pathway, EGR1
20	Tu et al. (2012) [49]	Taiwan	273	251/22	273/122	Tissue	NM	NM	246/27	NM	NM	TNM Stage (I, II, III and IV)	121 (I–III)	152 (IV)	T1, T2, T3, T4	N0, N+	Fadu, OECM−1, SAS, and 293FT	miR-149	Down-regulated	TaqMan qRT-PCR analysis (Applied Biosystems) (Carlsbad, CA, USA)	preliminary assays were unable to validate any gene
21	Jianbo Shi et al. (2019) [50]	China	260	99/71	170/90	Serum	adjuvant chemoradiotherapy	NM	68/102	67/103	NM	TNM Stage (I, II, III, and IV)	71	99	T1, T2, T3, T4	N0, N1, N2	Tumor-derived CRC lines	Serum miR-5100	Up-regulated	TaqMan miRNA Assays (Applied Biosystems)	NM
22	Jianbo Shi et al. (2019) [50]	China	260	99/71	170/90	Serum	adjuvant chemoradiotherapy	NM	68/102	67/103	NM	TNM Stage (I, II, III, and IV)	71	99	T1, T2, T3, T4	N0, N1, N2	Tumor-derived CRC lines	Serum miR-626	Up-regulated	TaqMan miRNA Assays (Applied Biosystems)	NM
23	Peng et al. (2014) [51]	R.O.C.	58	NM	29/0	Tissue	M4N treatment	NM	NM	NM	NM	p-stage III–IV	NM	43	T3, T4	N0, N+	OECM1, CG-C10, and SAS	miR-218,	Down-regulated	TaqMan miRNAs assay (ABI, Foster City, CA, USA)	Hedgehog and Wnt signaling cascades, SP1, MYC, and TP53 genes
24	Peng et al. (2014) [51]	R.O.C.	58	NM	29/0	Tissue	M4N treatment	NM	NM	NM	NM	p-stage III–IV	NM	43	T3, T4	N0, N+	OECM1, CG-C10, and SAS	Let-7g	Down-regulated	TaqMan miRNAs assay (ABI, Foster City, CA, USA)	Hedgehog and Wnt signaling cascades, SP1, MYC, and TP53 genes
25	Peng et al. (2014) [51]	R.O.C.	58	NM	29/0	Tissue	M4N treatment	NM	NM	NM	NM	p-stage III–IV	NM	43	T3, T4	N0, N+	OECM1, CG-C10, and SAS	miR-125b	Down-regulated	TaqMan miRNAs assay (ABI, Foster City, CA, USA)	Hedgehog and Wnt signaling cascades, SP1, MYC, and TP53 genes
26	Hikaru Nakashima et al. (2019) [52]	Japan	55	32/23	10-10	Plasma/Serum	5-fluorouracil (5-FU)–based CRT	SAS-R/CRR	NM	NM	Negative	TNM(I, IIa, IIb, III, and IV)	20	35	T2, T3, T4	N0, N1	SAS	miR-1290	Up-regulated	miScript II RT kit (QIAGEN, Hilden, Germany)	FOXC1/GLIPR1/BCL-2/NAT1
27	Ilyes Berania et al. (2017) [53]	Canada	58	41/17	58/36	NM	cetuximab	NM	42/16	24/34	13/45	NM	NM	NM	NM	NM	NM	miR-92a	Up-regulated	TaqMan MicroRNA Reverse Transcription Kit (Thermo FisherScientific).	PI3K/Akt/mTOR
28	Ilyes Berania et al. (2017) [53]	Canada	58	41/17	58/36	NM	cetuximab	NM	42/16	24/34	13/45	NM	NM	NM	NM	NM	NM	miR-548b	Up-regulated	TaqMan MicroRNA Reverse Transcription Kit (Thermo FisherScientific).	PI3K/Akt/mTOR
29	Ilyes Berania et al. (2017) [53]	Canada	58	41/17	58/36	NM	cetuximab	NM	42/16	24/34	13/45	NM	NM	NM	NM	NM	NM	miR-103	Up-regulated	TaqMan MicroRNA Reverse Transcription Kit (Thermo FisherScientific).	PI3K/Akt/mTOR
30	Ilyes Berania et al. (2017) [53]	Canada	58	41/17	58/36	NM	cetuximab	NM	42/16	24/34	13/45	NM	NM	NM	NM	NM	NM	miR-18a	Up-regulated	TaqMan MicroRNA Reverse Transcription Kit (Thermo FisherScientific).	PI3K/Akt/mTOR
31	Ilyes Berania et al. (2017) [53]	Canada	58	41/17	58/36	NM	cetuximab	NM	42/16	24/34	13/45	NM	NM	NM	NM	NM	NM	miR-205	Up-regulated	TaqMan MicroRNA Reverse Transcription Kit (Thermo FisherScientific).	PI3K/Akt/mTOR
32	Ilyes Berania et al. (2017) [53]	Canada	58	41/17	58/36	NM	cetuximab	NM	42/16	24/34	13/45	NM	NM	NM	NM	NM	NM	miR-532	Up-regulated	TaqMan MicroRNA Reverse Transcription Kit (Thermo FisherScientific).	PI3K/Akt/mTOR
33	Ilyes Berania et al. (2017) [53]	Canada	58	41/17	58/36	NM	cetuximab	NM	42/16	24/34	13/45	NM	NM	NM	NM	NM	NM	miR-20a	Up-regulated	TaqMan MicroRNA Reverse Transcription Kit (Thermo FisherScientific).	PI3K/Akt/mTOR
34	Ilyes Berania et al. (2017) [53]	Canada	58	41/17	58/36	NM	cetuximab	NM	42/16	24/34	13/45	NM	NM	NM	NM	NM	NM	miR-365	Up-regulated	TaqMan MicroRNA Reverse Transcription Kit (Thermo FisherScientific).	PI3K/Akt/mTOR
35	Yingying Zhao et al. (2020) [54]	China	90	54/36	90/13	NM	Cisplatin(DDP)	NP69	NM	NM	NM	I–II, III–IV	38	52	T1, T2, T3, T4	N0, N1, N2, N3	CNE-1, CNE-2, C666-1,5–8F and HONE-1	miR-1278	Down-regulated	PrimerScript RT-PCR Reagent Kit (TaKaRa, Dalian, China)	TGFβ pathway/ATG2B
36	M.K. Sannigrahi et al. (2017) [55]	India	110	87/23	279/0	Tissue	Cisplatin or 5-fluorouracil	UPCI:SCC-090 and SiHa	91/20	48/52	30/40	T(I, II, III, and IV)	70 (Stage II, III, and IV)		NM	NM	HPV+ (UPCI:SCC-090, CaSki, and SiHa) and HPV- (HEK-293, HaCat, and UPCI:SCC 116)	miR-139-3p	Down-regulated	NM	PDE2A
37	Yu-Chao Chang et al. (2015) [11]	China	135	NM	90/45	Tissue	Doxorubicin or cisplatin or 5-fluorouracil	NM	NM	NM	NM	T(I, II, III, and IV)	40	40	NM	NM	SAS/OECM-1/S-G	miR-494	Up-regulated	TaqMan miRNA assays with specific primer sets (Applied Biosystems, Carlsbad, CA, USA)	Bmi1/ADAM10
38	Bin Li et al. (2020) [56]	China	104	76/28	114/50	Tissue/Serum	Fluorouracil(5-FU)	KYSE150-FR	NM	NM	NM	Pathological Stages I and II, III and IV	76	28	T1, T2, T3, T4	N0, N1	KYSE150	miR-29c	Down-regulated	TaqMan human MicroRNA Low-Density Array Set	FBX031-p38 signaling

**Table 4 genes-13-02325-t004:** The genetic pathways involved in chemoresistance.

Down-Regulated			Up-Regulated		
Drug	miRNA	Pathway	Drug	miRNA	Pathway
CDDP-CRTX	miR-34a	TP53	CDDP-CRTX	miR-200b	Multidrug Resistance Gene
Cisplatin	miR-145	Akt pathway, MDR-1 gene, PPP2R1B Gene	MMC-CRTX	miR-155	Multidrug Resistance Gene
	miR-1278	TGFβ pathway/ATG2B		miR-146a	Multidrug Resistance Gene
	miR-139-3p	PDE2A	5-fluorouracil	miR-21	PI3K/AKT/S6 pathway, PTEN
				miR-1290	FOXC1/GLIPR1/BCL-2/NAT1
M4N Treatment	miR-218	Hedgehog and Wnt signaling cascades, SP1, MYC, and TP53 genes	Cisplatin, doxorubicin, paclitaxel	miR-196a	CDKN1B, ING5
	Let-7g	Hedgehog and Wnt signaling cascades, SP1, MYC, and TP53 genes	Cisplatin, docetaxel, 5-fluorouracil	miR-27a	FOXO1, MET, MDR-1 Gene
	miR-125b	Hedgehog and Wnt signaling cascades, SP1, MYC, and TP53 genes		miR-27b	FOXO1, MET, MDR-1 Gene
5-fluorouracil	miR-139-3p	PDE2A	Cisplatin	miR-200c	Akt pathway, MDR-1 gene, PPP2R1B Gene
	miR-29c	FBX031-p38 signaling		miR-21	Akt pathway, MDR-1 gene, PPP2R1B Gene
Others/NM	miR-200c	TP53	5-fluorouracil, leucovorin, oxaliplatin, and docetaxel	miRNA-194-5p	PTEN, BCL2, IGF1R, Wnt/β-catenin pathway; DKK2, CDH1, CD44, MYC, and ABCG2 expression
	miR-519d	CXCR4	Silibinin, doxorubicin, cisplatin, or fluorouracil	miRNA-494	ZEB2 and β-catenin signaling, ADAM10, FOXM1, CD44, and ALDH1
	miR-422a	FOXG1, CD73/NT5E oncogene, adenosine receptor-dependent signaling overexpressing CD73	Cisplatin and paclitaxel	miR-21	Inconclusive/NM
	miR-149	Inconclusive	Cetuximab	miR-15b-5p	p16, EGFR, and CD44; miR-15b-5p/TRIM-29/PTEN/AKT/mTOR signaling pathway
				miR-92a	PI3K/Akt/mTOR
				miR-548b	PI3K/Akt/mTOR
				miR-103	PI3K/Akt/mTOR
				miR-18a	PI3K/Akt/mTOR
				miR-205	PI3K/Akt/mTOR
				miR-532	PI3K/Akt/mTOR
				miR-20a	PI3K/Akt/mTOR
				miR-365	PI3K/Akt/mTOR
			Doxorubicin or cisplatin or 5-fluorouracil	miR-494	Bmi1/ADAM10
			Others/Nm	miR-675	Wnt signaling pathway, EGR1
				Serum miR-5100	NM
				Serum miR-626	NM

**Table 5 genes-13-02325-t005:** Publication bias indicators and hypothesis testing and heterogeneity testing analysis of miRNA-specific chemoresistance in HNC.

		**Heterogeneity Testing and Hypothesis Testing**															
	**Groups**				**Fixed**			**Mixed/Random**			**Hypothesis Test**						
		**Heterogeneity**			**HR**	**95% CI**		**HR**	**95% CI**		**Fixed Effects Model**			**Random Effects Model**			
		**Q**	** *p* **	**I^2^**		**Low**	**High**		**Low**	**High**	**Z**	** *p* **	**Studies**	**Z**	** *p* **		**Studies**
**1**	**Data from 2019–2021**	**118.56**	*0.00*	67.10	1.27	1.20	1.35	1.34	1.15	1.55	8.22	*0.00*	40	5.38	*0.00*		40
		**Publication Bias**															
	**Groups**	**Classic Fail-Safe N**			**Orwin Fail-Safe N**		**Begg and Mazumdar Test**				**Dual and Tweedie (Random Effects)**						
		**Z Value**		***p*-Value**	**HR in Observed**		**Tau**	**Z Value**		***p*-Value**	**Observed**		**Q Value**	**Adjusted**		**Q Value**	
1	Data from 2019–2021	8.63		*0.00*	1.30		0.16	1.43		*0.15*	1.52		118.57	1.34		145.70	

## Data Availability

Not applicable.

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
