# Peer review of "Clinical Investigation of Chemotherapeutic Resistance and miRNA Expressions in Head and Neck Cancers: A Thorough PRISMA Compliant Systematic Review and Comprehensive Meta-Analysis"

_genes, 2022, doi:10.3390/genes13122325_

Round 1

Reviewer 1 Report

In the manuscript by Jayaraj et al the authors summarize current knowledge on miRNA involvement in HNSCC chemoresistance. At first glance the study appears robust, however, upon closer examination many significant flaws are seen both in the study design, comprehensiveness, data presentation and analysis.

The manuscript needs thorough proofreading and language revision, ie. superflousus punctuation marks in headings, inconsistencies in headings or indentation. Errors in italic highlights.

While the design problems cannot really be addressed in a reasonable amount of time, some issues could at least make the manuscript more presentable.

P2L52 “observed in 2189 HNC cohorts” in the abstract is erroneous since results state “2189 people”

P2 L77 “The most common causes seem to be alcohol, smoking, and the high risk associated with HPV “ The sentence is ambiguous or erroneous. Possibly the authors refer to high-risk HPV types.

P4 L155 ambiguous “satisfied the inclusion criteria performed literature.”

P4 L165 Table 1 without proper bracketing of the search terms the table is meaningless and possibly suggests the search was not performed correctly.

Ie. first row entered in pubmed miRNA AND treatment OR "drug resistance" AND HNC OR "Head and Neck Cancer" gives 30907 publications while miRNA AND (treatment OR "drug resistance") AND (HNC OR "Head and Neck Cancer") returns 196. It is unlikely that the authors went through 30000 publications. It is also unclear how the results of different search strategies (rows in table 1) were combined

P5 L187 Why were studies done only on patients excluded? If the authors often highlight that the studies selected encompassed 2189 HNC patients, it is unusual that they opted to disregard purely clinical studies. On the other hand the authors rarely highlight how many cell lines were assessed yet this was an exclusion criteria.

P5 L187 Why were studies not discussing pathways excluded? In this way a lot of otherwise relevant data about miRNA is excluded and this review cannot be considered comprehensive

P5 L193 “ Studies whose full texts are not accessible were excluded.”  This needs to be further elaborated if significant number of studies was excluded in this way (apparently 41 of 113 studies identified at that step were excluded)

Why was the study search limited up to year 2021? Given that year 2022 is already at the end, additional studies might be missed. However this is trivial compared to previous exclusion steps that really make this study not comprehensive and warrant a complete rework of the data.

P7 L238 Figure 2 has errors and or overlap in the last 2 rows.

P8 L279 “The frequency of use of these cell lines” erroneously placed sentence and figure. This paragraph deals with studies in patients.

P9 Figure 4. It is not informative to state that 6 studies used “head and neck carcinoma cell line”. Why does Hacat cell line appear twice?

P9 L284 The section “Drug regulatory pathways …“ doesn’t really provide a clear link between the pathways noted on Figure 5 and drug resistance or sensitivity. One would expect some transport pathways or detoxification pathways to be drug resistance related while the pathways presented are very generic. Table 2 even specifies “multidrug resistance gene 1 pathway” which is absent from this figure

P11 L324 “The overall pooled effect estimate for the various miRNAs was 1.516 with a 95% confidence  interval of 1.303-1.765.” it is unclear whether this is an average effect of 34 miRNAs or how was an estimate made for 34 different targets

P11 L332 section on miRNA expression on survival is also unclear. Does this mean that 7 mirnas (15b,548b…) were consistently downregulated in 23 studies  and patients from those studies had improved survival?

P11 L 344 the paragraph on extent of variance is written in a highly detailed technical way that doesn’t fit well with the rest of the manuscript. It is unlikely that casual clinical or research readers interested in this topic will understand the statistical aspects of the reported values. Consider summarizing it in a more understandable way and move the highly specific technical reporting to the supplementary data. In either case reporting both T and T squared value is meaningless. Similarly holds for other sections dealing with the statistical validation of the dataset.

P11 section 3.6 the statement that publication bias was present was repeated twice in the paragraph. However the meaning of this bias and whether it affects the conclusions is unclear.

P12 table 2 should not be presented in the manuscript due to its size. It must be moved to the supplementary material and only a critical subset that can be made to fit A4 paper size (or reasonably spanning multiple pages) can be presented in the main text (PDF)

P20 and subsequent pages have wrong page/line  numbering.

Figure 6 seems to defeat the purpose of meta analysis concept. The point of a meta analysis or comprehensive review is to take results of previous studies regarding a specific topic and combine them to find additional insights beyond what is possible from a single study. However, Figure 1 presents data for a single study about most of the presented miRNAs. Ie miR15b-5p is found only once. The authors cannot really utilize the meta analysis design to assess a single study on this miR. Highlighting the “pool” of 2189 patients is also erroneous since only a small subset was actually assessed for any of those specifically listed mirs.

Author Response

Dear Reviewer,

Our sincere thanks for your comments which are helpful to enhance the quality of our paper. Much appreciated again.

Many Thanks

Prof Rama Jayaraj

Australia 

Reviewer 2 Report

In the present study, authors provided a systematic review and a meta-analysis with regards to chemotherapeutic resistance and miRNA expression and head and neck malignancies.

Although mechanistic behind these findings remain unclear, authors provided some novel and interesting results. The study can be considered for acceptance, after a major revision and the correction of multiple small tipos and mistakes throughout the manuscript (for example at 1.1 and 1.3 there is a point before the number, please correct all of these mistakes). As the whole paper is correctly written and data well-presented, I would suggest that authors correct all mistakes and provide further discussion on the possible underlying mechanistic associations.

Author Response

Reply to the reviewer: Dear Reviewer, thank you for the feedback. We have addressed the comments
mentioned.

Reviewer 3 Report

The manuscript is based on thorough research of the scientific literature and presents the most important data about the role of micro RNAs (miRNA) expression on the resistance to chemotherapy in head and neck cancers.

Starting from predefined search strategies, the authors extracted useful data from 23 bibliographic studies and used specific tests and indicators to perform a detailed meta-analysis. This study shows that of the 34 miRNAs considered, 22 showed increased expression, while 12 were under-expressed. This result was associated with an increased risk of death in the groups in which the miRNA was overexpressed. Among the 34 miRNAs that were investigated, the overexpression of the following miRNAs (miR15b-5p, miR-548b, miR-519d, miR-1278, miR-145, miR-200c, Hsa-miR139-3p) was associated with improved prognosis in head and neck cancers.

This study suggests the involvement of specific miRNAs as predictors of chemoresistance and sensitivity to therapy in HNC. In addition, the results presented in the current study emphasize the significance of miRNA expression as a therapeutic biomarker in medical oncology

 Thus, this review opens the way for a detailed meta-analysis to be performed on every patient to support and demonstrate the role of the expression of specific miRNAs in the management of the process of chemoresistance in HNC.

The conclusions are supported by the data presented, this paper being of interest considering the original approach on the very hot topic addressed.

Generally, the quality of the article is good, and the manuscript is interesting for the readers.

Overall, the English language and style need to be revised.

Author Response

Dear reviewer, Thank you for your comments addressing the paper. Revision of the manuscript has been done according to the comments.

Round 2

Reviewer 1 Report

The significant comments were not addressed. The manuscript is barely changed.

Regarding issue 4. "two wrongs dont make a right". Publishing a protocol paper with erroneous search strategy doesnt make it a correct strategy now.

Please understand that adding [topic] to the search strings doesnt solve anything and that a search for "miRNA" ... OR... "drug resistance"  will find all manuscripts on miRNA OR on drug resistance without selecting a subset where drug resistance was associated with miRNA. Such a search is completely meaningless and thus the string finds 30000 publications

saying that you performed the search in 2021 and that it changes anything is also not very credible and very easy to check.

Limiting the search in pubmed by date 1.1.2021 has still 26900 results.

Search: miRNA AND treatment OR "drug resistance" AND HNC OR "Head and Neck Cancer" Filters: from 1000/1/1 - 2021/1/1 Sort by: Most Recent

this simply demonstrates the search was either incorectly performed, performed by someone not familiar with such searches or incorectly reported. All cases are grounds for major revision which was not performed in this round of review.

It is strange that Peter Shaw, an expert on systematic reviews and the lead author of the protocol paper indicated by the authors is not an author of the final manuscript described by the same protocol paper

The manuscript also still doesnt clarify how different search strategies were combined

Issue 6 was similarly not addressed.

Criticism was that a large part of relevant publications were intentionally neglected. The answer was that the authors aimed to neglect relevant publications.

No justification why this in an apropriate way to aim to neglect studies in a systematic review is provided.

Issue 9 on why manuscripts without ful text data available were excluded was also not addressed.

THe answer to the reviewers "Reply to Reviewer: Only Studies whose full texts are able to provide a complete and accurate list of all data items that meet systematic review criteria. These differences have led to recommendations that full texts with multiple datasets will maximize the likelihood of finding relevant data to complete the qualitative synthesis (Systematic Review and MetaAnalysis)." is completely unclear and doesnt answer the question why the authors didnt approach authors and/or pay for access to those studies to include them in systematic review.

issue 15 is also not sufficiently addressed. The authors reply that Figure 6 supposedly depicts data on survival effects of the mirnas. However Figure 6 supposedly shows chemoresistance and not actual survival

Those terms while related cannot be used interchangeably and this leads to more questions about the  validity of the data altogether.

Issue 18. The table 2 splits across 8 pages and is completely uninformative due to terrible formatting. It remains so after the revision

Author Response

Dear Reviewer,

Please see the attached our response

Many Thanks

Prof Jayaraj

Australia
